# Cytokine and T cell responses in post-chikungunya viral arthritis: A cross-sectional study

Aileen Y. Chang[1]*, Sarah R. Tritsch[2], Carlos Andres Herrera Gomez[1], Liliana Encinales[3], Andres Cadena Bonfanti[4], Wendy Rosales[5], Evelyn Mendoza-Torres[5], Samuel Simmens[6], Richard L. Amdur[7], Christopher N. Mores[2], Paige Fierbaugh[1], Carlos Alberto Perez Hernandez[4], Geraldine Avendaño[4], Paula Bruges Silvera[4], Yerlenis Galvis Crespo[4], Alberto David Cabana Jimenez[4], Jennifer Carolina Martinez Zapata[4], Dennys Jimenez[4], Estefanie Osorio-Llanes[5], Jairo Castellar-Lopez[5], Karol Suchowiecki[1], Karen Martins[8], Melissa Gregory[9], Ivan Zuluaga[10], Abigale Proctor[2], Alfonso Sucerquia Hernández[1], Leandro Sierra[4], Maria Villanueva Colpas[4], Juan Carlos Perez Hernandez[4], Andres Alberto Figueroa Quast[4], Joaquin Andres Calderon De Barros[4], José Forero Mejía[4], Johan Penagos Ruiz[4], David Boyle[11], Gary S. Firestein[11]☯, Gary L. Simon[1]☯

1 Department of Medicine, George Washington University, Washington, DC, United States of America, 2 Department of Global Health, Milken Institute School of Public Health, George Washington University, Washington, DC, United States of America, 3 Department of Medicine, Allied Research Society, Barranquilla, Atlántico, Colombia, 4 Centro de Investigación, Clínica de la Costa SAS, Barranquilla, Atlántico, Colombia, 5 Advanced Biomedicine Research Group, Universidad Libre de Colombia, Seccional Barranquilla, Barranquilla, Atlántico, Colombia, 6 Department of Biostatistics and Bioinformatics, Milken Institute School of Public Health, George Washington University, Washington, DC, United States of America, 7 Feinstein Institute for Medical Research, Northwell Health, Manhasset, New York, United States of America, 8 U.S. Army Medical Research Institute of Infectious Diseases, Frederick, Maryland, United States of America, 9 Henry M. Jackson Foundation, In Support of Austere Environments Consortium for Enhanced Sepsis Outcomes (ACESO), Bethesda, Maryland, United States of America, 10 Universidad Libre de Barranquilla, Clínica Iberoamérica, Barranquilla, Atlántico, Colombia, 11 Department of Medicine, UC San Diego School of Medicine, San Diego, CA, United States of America

☯ These authors contributed equally to this work.
* chang@email.gwu.edu

## Abstract

### Objective

To define the relationship between chronic chikungunya post-viral arthritis disease severity, cytokine response and T cell subsets in order to identify potential targets for therapy.

### Methods

Participants with chikungunya arthritis were recruited from Colombia from 2019–2021. Arthritis disease severity was quantified using the Disease Activity Score-28 and an Arthritis-Flare Questionnaire adapted for chikungunya arthritis. Plasma cytokine concentrations (interleukin (IL)-1β, IL-2, IL-4, IL-6, IL-8, IL-10, IL-12p70, IL-13, interferon- and tumor necrosis factor (TNF)) were measured using a Meso Scale Diagnostics assay. Peripheral blood T cell subsets were measured using flow cytometry.

**Data Availability Statement:** All relevant data are within the manuscript and its Supporting Information files.

**Funding:** Research reported in this publication was supported by the National Institute of Arthritis and Musculoskeletal and Skin Diseases of the National Institutes of Health under Award Number K23AR076505 (niams. nih.gov) and the Pharmaceutical Research and Manufacturers of America Foundation (PhRMA) (www.phrmafoundation.org) Research Starter Grant in Translational Medicine and Therapeutics to AYC. The content is solely the responsibility of the authors and does not represent the official views of the National Institutes of Health nor the PhRMA Foundation. The funders had no role in study design, data collection and analysis, decision to publish, or preparation of the manuscript.

**Competing interests:** I have read the journal's policy and the authors of this manuscript have the following competing interests: AYC reports consulting for Valneva. RLA reports stock ownership of Abbvie, Pfizer, Johnson & Johnson, & Briston Myers Squibb. GSF reports consulting for Eli Lilly. Other authors report no commercial interest or conflicts of interest in relation to this work. This does not alter our adherence to PLOS ONE policies on sharing data and materials.

## Results

Among participants with chikungunya arthritis (N = 158), IL-2 levels and frequency of regulatory T cells (Tregs) were low. Increased arthritis disease activity was associated with higher levels of inflammatory cytokines (IL-6, TNF and CRP) and immunoregulatory cytokine IL-10 (p<0.05). Increased arthritis flare activity was associated with higher Treg frequencies (p<0.05) without affecting T effector (Teff) frequencies, Treg/Teff ratios and Treg subsets. Finally, elevated levels of IL-2 were correlated with increased Treg frequency, percent Tregs out of CD4$^+$ T cells, and Treg subsets expressing immunosuppressive markers, while also correlating with an increased percent Teff out of live lymphocytes (p<0.05).

## Conclusion

Chikungunya arthritis is characterized by increased inflammatory cytokines and deficient IL-2 and Treg responses. Greater levels of IL-2 were associated with improved Treg numbers and immunosuppressive markers. Future research may consider targeting these pathways for therapy.

## Introduction

Chikungunya arthritis is a debilitating condition responsible for significant morbidity(1) and loss of economic productivity [1,2]. Chikungunya virus (CHIKV) is an alphavirus spread by mosquitos. It affects an estimated one million people annually [3] and causes persistent arthritis in one-fourth of patients [4]. The arthritic potential of CHIKV is not unique; other alphaviruses such as Mayaro, Sinbis, Ross River, and O'nyong'nyong also cause severe arthritis [5]. There is currently no standard evidence-based treatment for alphavirus-induced arthritis. The objective of this study was to describe the peripheral blood cytokine and T cell profile of patients with chronic CHIKV arthritis in order to identify potential targets for therapy.

Alterations of regulatory T cell (Treg) populations may play a role in CHIKV arthritis pathogenesis. Tregs control the activation of CD4$^+$ T effector cells (Teff), and a lack of Tregs is associated with chronic CHIKV arthritis [6]. In addition, large numbers of Teff cells infiltrate the synovium in CHIKV arthritis [7,8], and the presence of Teffs is obligatory for the development of arthritis in mouse models [8]. The ratio of Treg to Teff cells is often established at the initial infection, affected by the quantity of antigen and duration of exposure to antigen. These factors drive T cell expansion and differentiation. For example, four or more days of initial viral symptoms predict persistent joint pain [4]. This longer duration of antigen exposure might affect T cell differentiation into Teffs vs. Tregs driving the finding that the frequency of Treg cells is lower in CHIKV arthritis patients than in recovered individuals [6]. Therefore, Tregs are a promising area of research to identify potential therapies for CHIKV arthritis.

Cytokines such as IL-2 are important modulators of T cell differentiation and a potential therapeutic target. In fact, low-dose IL-2 therapy is a preferential stimulator of Tregs vs. Teffs due to the presence of a unique high-sensitivity IL-2 receptor constitutively expressed only on Tregs [9]. Our Colombian cohort is one of the largest cohorts of CHIKV patients in the Americas to be followed longitudinally [4] and provides an ideal opportunity to further understand the immunology of viral arthritis and potential treatment targets. Our prior study revealed that low levels of IL-2 during acute infection were predictive of chronic joint pain [10]. Therefore, in this study our main goal was to define for the first time the complex relationship

between cytokine levels and T cell populations in varying levels of CHIKV arthritis disease activity in order to further understand potential therapeutic targets.

## Materials and methods

### Setting

Participants were recruited from Magdalena and Atlántico Departments, Colombia where a local CHIKV epidemic began in 2014. In Colombia, the first laboratory-confirmed autochthonous cases of CHIKV infection were reported in September 2014, leading to 19,435 CHIKV cases across the country from 2014 to 2016 [11].

### Participants

Participants age 18 and over with a history of CHIKV infection were analyzed. As per the Colombian Institute of Health, a clinically confirmed case of CHIKV infection is defined as a fever of >38°C, severe joint pain or arthritis, and the acute onset of erythema multiforme with symptoms not explained by other medical conditions. In addition, these individuals must reside in or have visited a municipality where evidence of CHIKV transmission is present or have traveled within 30 km of confirmed viral circulation. Participants were recruited from November 5, 2019 to August 25, 2021.

### Study design

Participants with clinically confirmed CHIKV infection were enrolled and followed as a part of the CHIKV cohort. Diagnosis of CHIKV was serologically confirmed via IgG antibody immunofluorescence (EUROIMMUN) as described below. Participants were excluded if a blood sample was not collected or CHIKV IgG was negative. An in-person history and physical were conducted to ascertain demographic characteristics, exposure history, and arthritis signs and symptoms. Blood samples were collected to determine levels of serum cytokines and T-cell subsets. Because of recruitment difficulties connected to the COVID-19 pandemic, the study design was shifted from a planned case-control design to the current Chikungunya case-only design. Patient enrollment continued until there were 158 patients with at least one study visit. An additional study visit was obtained for a subset of patients, but those are not included in this analysis.

### Ethics statement

The study protocol was approved by The George Washington University Institutional Review Board (Protocol: Colombian Arboviral Surveillance Protocol IRB #121611, GWU IRB, Washington, D.C., USA (FWA00005945) and the Clinica de la Costa IRB, Colombia (FWA IORG0008529)). Research on human subjects was conducted in compliance with regulations relating to the protection of human subjects and adhered to principles identified in the Belmont Report (1979). All data collection and research on human subjects for this publication were conducted under an IRB-approved protocol. All participants were adults and provided written informed consent during an in-person interview.

## Objectives and outcomes

The main objective of the study was to define the relationship between CHIKV arthritis disease severity, IL-2, and T-cell subsets in order to identify if arthritis therapies targeting enhanced regulatory T-cell activity may provide benefit.

### Ethnicity and gender reporting

Ethnicity and gender were self-reported with open-ended questions.

### Disease activity

Disease activity was assessed using the Disease Activity Score (DAS)-28 [12] with C-reactive protein (CRP), which includes assessment of the number of swollen and tender joints (out of the 28), CRP, and a visual analog global assessment of health. A DAS-28 of less than 2.6 indicates remission, 2.6–3.2 indicates low disease activity, 3.2–5.1 indicates moderate disease activity, and greater than 5.1 implies severe disease activity.

An arthritis flare is defined as the worsening of the arthritis disease process. A version of the Outcome Measures in Rheumatology Rheumatoid Arthritis-Flare Questionnaire (OMERACT-FQ) [13] was also adapted for use with CHIKV arthritis patients that contained five items to rate pain, physical function, stiffness, fatigue, and participation over the past week using 11-point numeric rating scales (0 = none to 10 = severe) where the composite score is the sum of the responses from the five domains ranging from 0 (no flare) to 50 (extreme flare). In general, flares are indicated by a total score >25, more than a week of symptoms, and patient classification of a "flare," but analysis as a continuous variable is recommended which was used in this study.

The Health Assessment Questionnaire (HAQ) Disability Index [14] was used to measure physical function, composed of a four-level difficulty scale for each item that is scored from 0 to 3, representing normal/no difficulty (0), some difficulty (1), much difficulty (2), and unable to do (3). There are 20 questions in eight categories of functioning–dressing, rising, eating, walking, hygiene, reach, grip, and usual activities. The value of the HAQ index can be interpreted in terms of three categories: mild difficulties to moderate disability (0–1), moderate to severe disability (1–2), and severe to very severe disability (2–3). Disability measured by the HAQ has repeatedly been correlated with mortality rates, progression of aging, and healthcare resource utilization.

Measures of stiffness and pain are of specific importance to quantifying CHIKV arthritis impact as they are associated with multiple domains of quality of life [15]. The Sparra Stiffness Questionnaire is a 12-item score calculated with a range of 0–15 to quantify the burden of stiffness. Pain intensity was assessed on a visual analog scale from 0–100 by both the participant and the physician.

Finally, the Patient-Reported Outcomes Measurement Information System (PROMIS) measures were used to assess quality of life [16]. PROMIS-29 was used to assess five domains (physical function, fatigue, anxiety, sleep disturbance and depression) with four questions for each domain. All domains were evaluated over the previous seven days except for physical function, which has no timeframe specified. Mobility was assessed with a separate four-item questionnaire.

### Sample collection and preparation

Blood was collected by venipuncture into K2EDTA vacutainers. The blood samples were centrifuged at room temperature (18–25°C) in a horizontal rotor for 20 minutes at 1,500 relative centrifugal force. Plasma was removed from the blood collection tubes and frozen at -80°C until analyzed. PMBCs were cryopreserved after a Ficoll separation.

### Cytokine levels

Multiplex assessment of a panel of plasma cytokines, including IL-1β, IL-2, IL-4, IL-6, IL-8, IL-10, IL-12p70, IL-13, interferon-γ and tumor necrosis factor (TNF), were measured using

the V-PLEX Proinflammatory Human Panel 1 Kit by Meso Scale Diagnostics according to the manufacturer's instructions.

## T-cell subsets

Flow cytometry was performed on a BD Celesta flow cytometer (BD Biosciences, San Jose, CA) and analyzed in FlowJo 10 (TreeStar Inc., Ashland, OR) to characterize **Tregs** defined as **CD3$^+$CD4$^+$CD25$^{hi/int+}$CD127$^-$FoxP3$^+$** and **Teff** defined as **CD3$^+$CD4$^+$CD25$^-$CD127$^-$**. All antibodies were titrated prior to use in this study. Peripheral blood mononuclear cells (PBMCs) were isolated from patients as described above and frozen until analysis. Once thawed, PBMCs were washed one time with DMEM containing FBS and once with 1X PBS. T cells were stained in two panels using antibodies against the following markers: CD3, CD4, CD25, CD28, CTLA4, CD127, FoxP3, GARP, Helios, HLA-DR, CD137 (4-1BB), CD45RA, and CCR7 (obtained from Biolegend, San Diego, CA). Live/Dead Fixable Aqua Dead Cell Stain and FoxP3 Fixation/Permeabilization Buffer were obtained from ThermoFisher Scientific (Waltham, MA).

## Data management

All patients were assigned a unique patient identification number, which was used in the database and for labeling patient samples. All patient data were free of personal identifiers and were stored in the REDCap database at The George Washington University.

## Chikungunya IgG immunofluorescence

An immunofluorescence-based assay was used to determine the presence or absence of anti-chikungunya virus IgG antibodies in enrolled patients (Euroimmun, Germany). Plasma samples were used to coat slides fitted with biochips containing chikungunya positive and negative cells. If present in the sample, the IgG antibody reacted with the positive cells and fluoresced. Slides were read using the 488nm excitation laser and a 4x objective on a Biotek Lionheart LX fluorescent microscope. This assay has a sensitivity of 97% and specificity of 96%.

## Statistical analysis

With 158 participants, there is 90% power to detect a Pearson correlation of .25 (alpha = .05, 2-sided), which is an approximation to the expected power to detect a similar Spearman rank-order correlation. Data was analyzed using SAS (version 9.4) and R (version 4.3.1). All data distributions for continuous variables were first assessed for outliers and non-normality by examining plots. Many of the laboratory-based variables were highly skewed and some had outliers (determined visually) or values below the limit of detection. The outliers were all consistent with standard curves and therefore assumed to be valid non-missing values and retained in analyses. Descriptive statistics for continuous variables are presented as medians with first and 3$^{rd}$ quartiles. Bivariable associations were examined for linearity through scatterplots. Missing data were not imputed. Because most distributions were non-normal, all significance testing was through non-parametric tests (Spearman Rank-order correlations [$r_s$] and Kruskal-Wallis tests using the Chi-Square approximation). The threshold for significance was alpha <0.05 using 2-sided tests. Because the study aim is to identify all markers with a *possible* association with disease activity, rather than to confirm previous findings, p-values were unadjusted for multiple testing. Lack of adjustment can result in more Type I statistical errors (false positive results), but greater power and therefore fewer Type II errors (false negative findings). Also, to maximize power, raw scores, rather than categorized scores, for the DAS-28 were used in significance tests.

## Results

### Gender and disease activity in the Colombian cohort

One hundred seventy-five participants with clinical CHIKV infection at least three months prior were enrolled between November 2019 and August 2021. Seventeen patients were excluded due to a lack of serologic confirmation of CHIKV infection by IgG immunofluorescence. One hundred fifty-eight participants were included in this analysis. The raw data is included as a supplementary dataset.

All participants self-identified as the Colombian mestizo ethnicity (i.e., mixed European, often Iberian, and indigenous Latin American ancestry). The demographic characteristics of the participants with serologically confirmed CHIKV and classified according to Disease Activity Score (DAS)-28 are shown in Table 1. Older participants showed a trend towards greater disease activity that was not significant ($r_s$ = .14; p = 0.07). The majority of the participants were female, and females had more active arthritis than males (Kruskal-Wallis $\chi^2(1)$ = 4.8, p = 0.03). The majority of the study population had at least a secondary school education. Educational level was not related to disease activity.

### Disability, pain, stiffness and decreased quality of life in chikungunya arthritis

Disease Activity Scores (DAS-28) ranged from 0.96 (remission) to 6.48 (severe arthritis activity). Participants reported moderate disease activity on average, with a DAS-28 mean of 3.66 (SD = 1.23). The characteristics of CHIKV arthritis according to DAS-28 are shown in Table 2. A greater number of days of fever during the initial CHIKV infection correlated with increased disease activity. More severe disease activity correlated with higher flare scores and greater disability, patient-reported and physician-reported pain intensity, and stiffness. Higher disease activity also correlated with effects on patient-reported physical function, fatigue, anxiety, sleep and mobility. Patients in remission had lower depression scores, although the overall association of depression with CHIKV arthritis disease activity was not statistically significant.

### Treatment: Acetaminophen, non-steroidal anti-inflammatories (NSAIDS) and steroids with minimal use of immunomodulatory medications

Medication use by participants with CHIKV arthritis is shown in Table 3. Acetaminophen (90.2%) and ibuprofen (74.0%) were the most commonly used arthritis therapies. Ibuprofen use was most prevalent in participants with lesser arthritis activity. Steroids were used in approximately one third of participants, with the highest use in severe disease activity. Methotrexate was only used by one participant with moderate arthritis disease severity. Fifteen percent of participants reported the use of other arthritis therapies, including the NSAID meloxicam, the muscle relaxant methocarbamol, and physical therapy.

**Table 1. Demographic characteristics of participants of chikungunya arthritis cases grouped by disease activity Score-28.**

|  | All Cases | Arthritis Disease Activity | | | |
|---|---|---|---|---|---|
|  |  | Remission 0–2.6 | Mild 2.6–3.2 | Moderate 3.2–5.1 | Severe 5.1+ |
| Age (mean, SD) | 48.6 (15.9) | 42.6 (18.6) | 48.7 (18.9) | 49.6 (14.8) | 52.9 (10.1) |
| % Female | 80% (127/158) | 61% (17/28) | 95% (20/21) | 82% (75/92) | 88% (15/17) |
| % With at least secondary school education | 63% (99/157) | 64% (18/28) | 67% (14/21) | 65% (59/91) | 47% (8/17) |

**Table 2. Associations of chikungunya arthritis characteristics (mean, SD) grouped by disease activity Score-28 and compared using Spearman's correlation coefficient.**

| | All Cases | Remission 0–2.6 | Mild 2.6–3.2 | Moderate 3.2–5.1 | Severe 5.1+ | *rs* | *p* |
|---|---|---|---|---|---|---|---|
| **Arthritis Disease Activity Measured by the Disease Activity Score-28** | | | | | | | |
| **Days of fever during initial chikungunya infection** | 5.5 (4.7) | 4.0 (3.7) | 4.2 (1.9) | 5.8 (5.1) | 7.4 (5.2) | 0.24 | 0.002 |
| Flare score, 0–50 | 25.6 (12.3) | 10.4 (12.6) | 27.4 (8.4) | 28.3 (10.2) | 33.8 (5.3) | 0.57 | <0.001 |
| **Health Assessment Questionnaire score, 0–3** | 0.78 (0.57) | 0.3 (0.4) | 0.8 (0.6) | 0.9 (0.6) | 1.0 (0.4) | 0.47 | <0.001 |
| **Pain intensity, patient-rated, 0–100** | 61.9 (27.1) | 24.0 (28.6) | 60.0 (21.2) | 71.2 (17.9) | 76.8 (12.9) | 0.56 | <0.001 |
| **Pain intensity, physician-rated, 0–100** | 48.0 (28.4) | 16.1 (20.6) | 50.9 (18.4) | 54.9 (25.2) | 59.0 (31.0) | 0.51 | <0.001 |
| **Sparra Stiffness Score, 0–15** | 5.7 (4.0) | 2.0 (3.0) | 5.6 (2.8) | 6.4 (3.7) | 8.7 (4.0) | 0.55 | <0.001 |
| **Quality of Life Measured by the Patient Reported Outcomes Measurement Information System (PROMIS)** | | | | | | | |
| **Physical function** | 45.3 (8.2) | 52.8 (6.9) | 46.8 (9.0) | 43.8 (7.3) | 38.8 (4.5) | -0.49 | <0.001 |
| **Fatigue** | 54.6 (9.9) | 45.9 (10.7) | 55.6 (7.4) | 56.1 (8.9) | 60.1 (7.8) | 0.38 | <0.001 |
| **Anxiety** | 57.0 (8.7) | 51.5 (9.7) | 57.1 (7.9) | 57.9 (8.3) | 60.9 (6.3) | 0.30 | <0.001 |
| **Sleep disturbance** | 50.5 (8.7) | 47.0 (8.2) | 49.4 (9.5) | 50.8 (8.6) | 56.0 (7.2) | 0.31 | <0.001 |
| **Depression** | 52.7 (9.0) | 49.7 (8.1) | 55.6 (9.4) | 52.8 (9.1) | 53.5 (8.5) | 0.10 | 0.10 |
| **Mobility** | 43.3 (9.6) | 53.5 (9.5) | 42.6 (9.8) | 41.4 (8.2) | 37.4 (2.8) | -0.47 | <0.001 |

$r_s$ denotes the Spearman rank correlation coefficient.

Note: N = 158 except for Sparra Stiffness score N = 154.

## Inciting factors for arthritis flares

The causes of arthritis flares in CHIKV arthritis cases are shown in Table 4. Exercise (42%) and a recent infection (18%) were most commonly associated with arthritis flares. Participants with increasing arthritis severity were more likely to report recent infection as a cause of flare (H(1) = 5.89, P = 0.02). A few participants reported specific foods or new medications as inciting factors for arthritis flares. Many participants reported other factors that contributed to arthritis flares, including exposure to cold temperatures and any activity that required the participant to get into an unusual or specific body position.

## Cytokines and chikungunya arthritis disease activity

Among participants with post-CHIKV viral arthritis, IL-2 levels were a median of 0.13 pg/ml with a range = 0–3.66 pg/ml. The correlations between C-reactive protein, cytokine levels and Treg subsets with arthritis disease severity are shown in Table 5. Disease severity was graded using the DAS-28 and the flare score. C-reactive protein correlated with both DAS-28 arthritis disease activity ($r_s$(155) = 0.36, p = <0.001) and increased flare severity ($r_s$(155) = 0.17, p = 0.03). Increased arthritis disease activity was associated with higher levels of inflammatory cytokines, including IL-6 ($r_s$(156) = 0.24, p = 0.03) and TNF ($r_s$(156) = 0.21, p = 0.009) and immunoregulatory cytokine IL-10 ($r_s$(156) = 0.17, p = 0.03). Levels of IL-1β, IL-2, IL-4, IL-8, IL-12p70, IL-13 and interferon-γ were not correlated with disease severity.

## Treg abundance and chikungunya arthritis flare

Among participants with post-CHIKV viral arthritis, the median percent Tregs out of CD4[+] lymphocytes was low (1.79%, range 0.11–8.15%). Increased arthritis disease flare activity was

**Table 3. Percentage of chikungunya arthritis cases currently using various arthritis therapies grouped by disease activity Score-28 and compared using Kruskal-Wallis analysis.**

| Medication | All Cases | Remission 0–2.6 | Mild 2.6–3.2 | Moderate 3.2–5.1 | Severe 5.1+ | H (1 df) | P |
|---|---|---|---|---|---|---|---|
| Acetaminophen | 90% (139/154) | 70% (19/27) | 100% (20/20) | 96% (86/90) | 82% (14/17) | 1.85 | 0.17 |
| Ibuprofen | 74% (115/155) | 46% (13/28) | 95% (19/20) | 78% (70/90) | 76% (13/17) | 4.64 | 0.03 |
| Steroids | 33% (50/153) | 19% (5/27) | 20% (4/20) | 37% (33/89) | 47% (8/17) | 13.48 | <0.001 |
| Aspirin | 25% (38/154) | 18% (5/28) | 25% (5/20) | 26% (23/89) | 29% (5/17) | 0.62 | 0.43 |
| Methotrexate | 1% (1/151) | 0% (0/27) | 0% (0/20) | 1% (1/87) | 0% (0/17) | 0.88 | 0.35 |
| Other therapy | 15% (23/154) | 7% (2/27) | 25% (5/20) | 17% (15/90) | 6% (1/17) | 0.27 | 0.87 |

associated with higher levels of Tregs (percent Treg/live lymphocytes) ($r_s$(156) = 0.17, p = 0.03) without an effect on Teff, Treg/Teff ratios, and Treg subsets (Table 5).

## Association between IL-2 and Treg numbers and expression of immunosuppressive markers

The association between IL-2 and T cell subsets is shown in Table 6. Higher IL-2 levels were correlated with increased Treg counts ($r_s$(156) = 0.17, p = 0.03), percent Tregs out of CD4$^+$ T cells ($r_s$(156) = 0.20, p = 0.01), and immunosuppressive Treg markers, including CTLA4 ($r_s$(156) = 0.27, p = <0.001), Helios ($r_s$(156) = 0.17, p = 0.03) and HLA-DR ($r_s$(156) = 0.18, p = 0.03), and Treg activation marker CD45RA ($r_s$(156) = 0.18, p = 0.02). Higher IL-2 also correlated with lower CCR7 expression ($r_s$(156) = -0.21, p = 0.007) [17]. Higher IL-2 was also associated with an increased percent Teff out of live lymphocytes ($r_s$(156) = 0.21, p = 0.008).

## Discussion

Our primary findings were that post-CHIKV viral arthritis disease activity was associated with increased inflammatory cytokines and IL-10 immunoregulatory cytokine concentrations. Arthritis disease flare was weakly associated with an increase in Treg percentage of live lymphocytes without an increase in Treg activation markers. Lastly, higher IL-2 levels correlated with increased Treg number and functional markers, but also increased Teff percentages, suggesting that further study into the IL-2 pathway for therapeutic intervention may be warranted.

**Table 4. Causes of arthritis flares in chikungunya arthritis cases grouped by disease activity Score-28 compared using Kruskal-Wallis analysis.**

| Cause | All Cases | Remission 0–2.6 | Mild 2.6–3.2 | Moderate 3.2–5.1 | Severe 5.1+ | H (df = 1) | P |
|---|---|---|---|---|---|---|---|
| Exercise | 42% (66/158) | 32% (9/28) | 24% (5/21) | 50% (46/92) | 35% (6/17) | 1.27 | 0.26 |
| Infection | 18% (29/158) | 7% (2/28) | 14% (3/21) | 22% (20/92) | 24% (4/17) | 5.89 | 0.02 |
| Food | 4% (6/158) | 0% (0/28) | 5% (1/21) | 5% (5/92) | 0% (0/17) | 0.07 | 0.79 |
| Medications | 1% (1/158) | 0% (0/28) | 0% (0/21) | 1% (1/92) | 0% (0/17) | 0.05 | 0.82 |
| Other factor | 16% (25/158) | 14% (4/28) | 29% (6/21) | 15% (14/92) | 6% (1/17) | 2.39 | 0.12 |

**Table 5. Associations of cytokines and T cell subsets in chikungunya arthritis participants with arthritis disease activity and the flare score.**

| | All Cases | Disease Activity Score-28 *Median (Q1–Q3)* | | | | Correlation with DAS-28 Score | | Correlation with Flare score | |
|---|---|---|---|---|---|---|---|---|---|
| | | Remission 0–2.6 | Mild 2.6–3.2 | Moderate 3.2–5.1 | Severe 5.1+ | rs | p-value | rs | p-value |
| C-Reactive Protein | 1.59 (0–31.5) | 0 (0–2.13) | 0.95 (0.39–2.40) | 1.91 (0.71–4.35) | 3.55 (1.61–4.58) | 0.36 | <0.001 | 0.17 | 0.03 |
| IL-1β | 0 (0–1.47) | 0 (0–0) | 0 (0–0.03) | 0 (0–0.04) | 0 (0–0.03) | 0.07 | 0.39 | 0.03 | 0.67 |
| IL-2 | 0.13 (0–3.66) | 0.12 (0–0.26) | 0.17 (0–0.23) | 0.13 (0.01–0.22) | 0.22 (0–0.30) | 0.05 | 0.50 | -0.01 | 0.93 |
| IL-4 | 0.05 (0–0.52) | 0.04 (0.02–0.06) | 0.05 (0.04–0.13) | 0.04 (0.03–0.08) | 0.08 (0.04–0.26) | 0.10 | 0.19 | 0.06 | 0.47 |
| IL-6 | 0.25 (0–37.35) | .15 (0.11–0.32) | 0.29 (0.16–0.47) | 0.26 (0.16–0.63) | 0.32 (0.22–1.15) | 0.24 | 0.03 | 0.15 | 0.06 |
| IL-8 | 1.21 (0.11–17.72) | 0.77 (0.45–2.21) | 1.35 (1.03–3.75) | 1.22 (0.48–3.78) | 1.65 (0.62–4.41) | 0.09 | 0.28 | 0.15 | 0.07 |
| IL-10 | 0.10 (0.02–8.34) | 0.09 (0.05–0.15) | 0.10 (0.09–0.44) | 0.10 (0.07–0.26) | 0.28 (0.10–0.06) | 0.17 | 0.03 | 0.14 | 0.09 |
| IL-12p70 | 0.19 (0–50.03) | 0.16 (0.12–0.25) | 0.23 (0.12–0.28) | 0.18 (0.10–0.32) | 0.23 (0.13–0.39) | 0.04 | 0.61 | 0.00 | 0.95 |
| IL-13 | 1.3 (0–9.68) | 1.35 (0.98–1.61) | 1.49 (1.31–3.76) | 1.33 (0.97–1.86) | 1.46 (1.13–5.88) | 0.07 | 0.38 | 0.02 | 0.80 |
| IFN-γ | 1.73 (0.21–15.30) | 1.03 (0.59–2.88) | 2.80 (2.09–4.13) | 1.69 (0.81–2.80) | 2.10 (0.92–3.22) | 0.10 | 0.33 | 0.07 | 0.39 |
| TNF | 0.57 (0.10–6.08) | 0.35 (0.27–0.67) | 0.62 (0.48–2.11) | 0.59 (0.39–1.44) | 0.85 (0.40–2.89) | 0.21 | 0.009 | 0.16 | 0.04 |
| % Teff /Live lymphocytes | 2.04 (0.02–10.00) | 1.85% (0.94–2.78) | 1.77% (0.90–2.33) | 2.07% (1.27–3.38) | 2.28% (1.91–3.66) | 0.14 | 0.08 | -0.20 | 0.85 |
| %Teff/CD4+ T cells | 0.08 (0–4.29) | 0.08% (0.04–0.18) | 0.06% (0.04–0.09) | 0.08% (0.05–0.16) | 0.08% (0.05–0.11) | 0.02 | 0.84 | -0.02 | 0.84 |
| Treg/Teff count ratio | 0.15 (0–12.37) | 0.13 (0.03–0.33) | 0.26 (0.14/0.51) | 0.14 (0.06–0.30) | 0.17 (0.13–0.45) | 0.04 | 0.62 | 0.12 | 0.12 |
| %Treg/Live lymphocytes | 0.22 (0.02–1.57) | 0.16 (0.07–0.29) | 0.30 (0.18–0.49) | 0.24 (0.12–0.39) | 0.23 (0.18–0.38) | 0.10 | 0.19 | 0.17 | 0.03 |
| % Treg/CD4+ cells | 1.79 (0.11–8.15) | 1.36 (0.43–2.31) | 2.00 (1.47–2.58) | 1.59 (0.78–2.43) | 1.92 (1.25–2.56) | 0.12 | 0.14 | 0.15 | 0.06 |
| Tregs CTLA4+ | 50 (0–680) | 13 (4–85) | 105 (38–145) | 41 (13–106) | 88 (54–128) | 0.11 | 0.17 | 0.14 | 0.07 |
| Tregs GARP+ | 0 (0–5) | 0 (0–1) | 0 (0–1) | 0 (0–1) | 0 (0–1) | 0.00 | 0.99 | 0.00 | 0.97 |
| Tregs Helios+ | 231 (1–2,691) | 105 (61–479) | 495 (181–1,161) | 22 (124–534) | 386 (160–703) | 0.08 | 0.29 | 0.12 | 0.11 |
| Tregs HLADR+ | 121 (0–1,429) | 63 (27–199) | 218 (127–474) | 105 (55–260) | 159 (89–311) | 0.08 | 0.31 | 0.13 | 0.11 |
| Tregs 41BB+ | 6 (0–92) | 4 (2–9) | 13 (5–18) | 6 (2–11) | 7 (2–17) | -0.03 | 0.70 | 0.08 | 0.31 |
| Tregs CCR7+ | 15 (0–260) | 22 (8–58) | 15 (6–71) | 15 (4–42) | 5 (3–29) | -0.14 | 0.07 | -0.07 | 0.34 |
| Tregs CD28+ | 323 (2–2,414) | 269 (115–405) | 507 (283–983) | 303 (149–578) | 371 (210–539) | 0.00 | 0.97 | 0.05 | 0.51 |
| Tregs CD45RA+ | 61 (0–874) | 30 (12–96) | 97 (52–371) | 52 (21–164) | 69 (55–126) | 0.08 | 0.35 | 0.12 | 0.13 |

Note: N = 158, except C-Reactive Protein N = 157.

Cytokine concentrations are shown in pg/ml.

$r_s$ denotes the Spearman rank correlation efficien.

Table 6.  Associations of T cell subsets with IL-2 level in chikungunya arthritis cases.

| T cell subset | Spearman correlation (rs) | P value |
|---|---|---|
| Teff count | 0.14 | 0.07 |
| % Teff /Live lymphocytes | 0.21 | 0.008 |
| % Teff/CD4+ T cells | 0.12 | 0.13 |
| Treg count | 0.17 | 0.03 |
| %Treg/Live lymphocytes | 0.15 | 0.053 |
| % Treg/CD4+ T cells | 0.20 | 0.01 |
| Tregs CTLA4+ | 0.27 | <0.001 |
| Tregs Helios | 0.17 | 0.03 |
| Tregs HLADR | 0.18 | 0.03 |
| Tregs CCR7 | -0.21 | 0.007 |
| Tregs CD45RA | 0.18* | 0.02 |

Note: N = 158.

To our knowledge, this is the first report to describe in detail the characteristics of post-CHIKV arthritis approximately five years after the initial infection. Consistent with prior studies, CHIKV arthritis disease severity was associated with older age and female sex [18]. On average, participants reported moderate disease activity consistent with a prior study [19]. A greater number of days of fever during initial chikungunya infection correlated with increased disease activity consistent with a previous study [4], suggesting that prolonged exposure to active CHIKV infection contributes to persistent arthritis. More severe disease activity was correlated with greater disability, patient- and physician-reported pain intensity, and stiffness. Higher disease activity was also correlated with effects on patient-reported physical function, fatigue, anxiety, sleep and mobility, demonstrating the impact of CHIKV arthritis on most major domains of quality of life.

IL-2 levels in this post-CHIKV arthritis cohort were low in comparison to healthy adults [20] and rheumatoid arthritis [21], which is also associated with deficient IL-2 production [22]. In addition, we also reported low Treg out of CD4 lymphocyte percentages in post-CHIKV arthritis. This is consistent with the findings of Kulkarni et al., who showed that the frequency of Treg cells was lower in acute and chronic CHIKV arthritis patients than in recovered individuals, healthy controls and rheumatoid arthritis patients [6]. The lack of IL-2 and Tregs in post-CHIKV disease may result from prolonged exposure to the chikungunya virus and viral antigens that drive differentiation of naïve T cells into Teff as opposed to Tregs. This hypothesis is consistent with our findings that four or more days of initial viral symptoms was a significant predictor of persistent joint pain [4] and with our finding in this cohort that longer days of initial fever were associated with more arthritis disease severity years after initial infection.

CHIKV arthritis disease severity was not associated with IL-2 levels in our study, consistent with the findings from Kelvin et al. and Ng et al. [23,24]. However, we did find disease activity was associated with inflammatory cytokines IL-6, TNF and CRP, and anti-inflammatory cytokine IL-10. Similar findings have been reported with chronic CHIKV arthritis reported with IL-6 [25–27], TNF [27], CRP [28], and IL-10 [27]. These findings are also consistent with the few other studies examining CHIKV arthritis disease severity that also report higher disease severity associated with increased levels of IL-6 [24,29]. Chronic CHIKV arthritis is characterized by significantly higher IL-6 than found in recovered patients [26]. The cytokine/chemokine milieu suggests that disease severity in post-CHIKV arthritis is driven by processes

such as IL-6-mediated inflammation and TNF-mediated T-cell chemotaxis, in addition to an immunomodulatory response from IL-10 in an attempt to regulate immune-related damage. This suggests that therapies targeting inflammatory cytokines such as IL-6 and TNF may be fruitful areas of CHIKV arthritis research.

Our finding that increased arthritis flare was mildly associated with an increased percent Tregs out of live lymphocytes and that increasing IL-2 was associated with gains in Treg frequency and immunomodulatory markers suggests a potential opportunity for low-dose IL-2 as a novel therapy to target Tregs for post-CHIKV Treg deficiency. Furthermore, the ratio of Treg/Teff was unchanged, with increasing disease activity and flare. We did not find increases in the expression of other markers important in Treg suppressive function with increased disease activity, such as markers related to Treg immunosuppressive function (CTLA4) [30], differentiation (GARP) [31], cell function (Helios, CCR7 and CD28) [17,32,33], migration (where a CCR7 deficiency promotes Treg accumulation in inflamed tissue [17], activation (HLA-DR+ CD45RA) [30], and proliferation (4-1BB) [34]. Our findings suggest that even with increasing arthritis disease activity, CHIKV arthritis cases display a limited Treg response.

Similar to our findings, Macêdo Gois et al. [35] described the association between CHIKV infection and a reduced Tregs frequency, along with the impaired expression and production of Treg functional markers, including CD39, CD73, perforin, granzyme, programmed death 1 (PD-1), transforming growth factor (TGF)-β and CTLA4, in both acute and chronic phases of the disease. This suggests that Treg cells possess a poor regulatory capacity after CHIKV infection. Our studies indicate that boosting Treg responses may be a therapeutic target in treating CHIKV arthritis flares.

We also found that higher IL-2 levels were associated with an increased percent Teff out of live lymphocytes. This reflects the pleiotropic effects of IL-2 and the importance of identifying the optimal dose of low-dose IL-2 therapy in human trials of CHIKV arthritis to boost Tregs and Treg/Teff ratios without enhancing Teff cells. In therapeutic cancer trials, high-dose IL-2 (>50 Million International Units (MIU)/8 hours) [36] stimulates Teffs to attack cancer cells [37], whereas in low-dose IL-2 therapy (<6 MIU/day) [36] for autoimmune diseases, Tregs are stimulated [38]. The ability of low-dose IL-2 to preferentially stimulate Tregs rather than Teffs is due to the constitutive expression of a high-sensitivity IL-2 receptor [9], a 10–20-fold lower activation threshold for IL-2 compared to Teff, and gene activation that requires IL-2 doses that are 100-times lower in Tregs compared to Teffs [39]. Therefore, significantly lower amounts of recombinant IL-2 may be used in the treatment of autoimmune diseases such as post-CHIKV chronic arthritis compared to dosages used in cancer therapy; however, further trials are needed to clarify the optimal low dose in CHIKV arthritis.

The limitations of this study include the cross-sectional design that did not capture the cytokine and T cell profile during the week prior to the evaluation, which may have contributed to the development of arthritis flares. In addition, significantly higher Treg percentages were appreciated in the synovial tissue of rheumatoid arthritis patients compared to peripheral blood [40], and synovial Treg numbers increase in parallel with worsening inflammation [41,42]. Thus, synovial studies in CHIKV arthritis patients would be useful. Finally, additional healthy controls from this population would have been useful for comparison.

## Conclusions

In conclusion, our study shows that CHIKV arthritis is characterized by increased inflammatory cytokines but deficient Treg responses, suggesting that arthritis therapies targeting enhanced regulatory Treg activity might be a plausible focus for further investigation.

## Supporting information

**S1 Dataset. Observational study dataset.** Raw data from the observational study.
(XLSX)

## Acknowledgments

Allied Research Society and the Chikungunya Arthritis Mechanisms in the Americas cohort participants who have dedicated their time.

## Author contributions

**Conceptualization:** Aileen Y. Chang, Sarah R. Tritsch, Liliana Encinales, Evelyn Mendoza-Torres, Samuel Simmens, Karen Martins, Gary S. Firestein, Gary L. Simon.

**Data curation:** Aileen Y. Chang, Sarah R. Tritsch, Carlos Andres Herrera Gomez, Andres Cadena Bonfanti, Wendy Rosales, Samuel Simmens, Richard L. Amdur, Paige Fierbaugh, Carlos Alberto Perez Hernandez, Geraldine Avendaño, Paula Bruges Silvera, Yerlenis Galvis Crespo, Alberto David Cabana Jimenez, Jennifer Carolina Martinez Zapata, Estefanie Osorio-Llanes, Jairo Castellar-Lopez, Karol Suchowiecki, Melissa Gregory, Abigale Proctor, Alfonso Sucerquia Hernández, Leandro Sierra, Maria Villanueva Colpas, Juan Carlos Perez Hernandez, Andres Alberto Figueroa Quast, Joaquin Andres Calderon De Barros, José Forero Mejía, Johan Penagos Ruiz.

**Formal analysis:** Aileen Y. Chang, Sarah R. Tritsch, Wendy Rosales, Evelyn Mendoza-Torres, Samuel Simmens, Richard L. Amdur, Karen Martins, Melissa Gregory, Abigale Proctor, José Forero Mejía, David Boyle, Gary S. Firestein, Gary L. Simon.

**Funding acquisition:** Aileen Y. Chang.

**Investigation:** Aileen Y. Chang, Sarah R. Tritsch, Carlos Andres Herrera Gomez, Liliana Encinales, Andres Cadena Bonfanti, Wendy Rosales, Evelyn Mendoza-Torres, Christopher N. Mores, Paige Fierbaugh, Carlos Alberto Perez Hernandez, Geraldine Avendaño, Paula Bruges Silvera, Yerlenis Galvis Crespo, Alberto David Cabana Jimenez, Jennifer Carolina Martinez Zapata, Dennys Jimenez, Estefanie Osorio-Llanes, Jairo Castellar-Lopez, Karol Suchowiecki, Karen Martins, Melissa Gregory, Ivan Zuluaga, Alfonso Sucerquia Hernández, Leandro Sierra, Maria Villanueva Colpas, Juan Carlos Perez Hernandez, Andres Alberto Figueroa Quast, Joaquin Andres Calderon De Barros, José Forero Mejía, Johan Penagos Ruiz, Gary S. Firestein, Gary L. Simon.

**Methodology:** Aileen Y. Chang, Sarah R. Tritsch, Carlos Andres Herrera Gomez, Liliana Encinales, Andres Cadena Bonfanti, Wendy Rosales, Evelyn Mendoza-Torres, Samuel Simmens, Richard L. Amdur, Christopher N. Mores, Geraldine Avendaño, Dennys Jimenez, Estefanie Osorio-Llanes, Jairo Castellar-Lopez, Karol Suchowiecki, Karen Martins, Melissa Gregory, Alfonso Sucerquia Hernández, José Forero Mejía, David Boyle, Gary S. Firestein, Gary L. Simon.

**Project administration:** Aileen Y. Chang, Sarah R. Tritsch, Carlos Andres Herrera Gomez, Liliana Encinales, Andres Cadena Bonfanti, Wendy Rosales, Evelyn Mendoza-Torres, Christopher N. Mores, Paige Fierbaugh, Geraldine Avendaño, Paula Bruges Silvera, Dennys Jimenez, Karol Suchowiecki, Melissa Gregory, Ivan Zuluaga, Alfonso Sucerquia Hernández, José Forero Mejía, Gary L. Simon.

**Resources:** Aileen Y. Chang, Andres Cadena Bonfanti, Wendy Rosales, Evelyn Mendoza-Torres, Christopher N. Mores, Melissa Gregory, David Boyle, Gary S. Firestein, Gary L. Simon.

**Software:** Aileen Y. Chang, Sarah R. Tritsch, Samuel Simmens, Richard L. Amdur, Christopher N. Mores, José Forero Mejía.

**Supervision:** Aileen Y. Chang, Carlos Andres Herrera Gomez, Liliana Encinales, Andres Cadena Bonfanti, Wendy Rosales, Evelyn Mendoza-Torres, Christopher N. Mores, Paula Bruges Silvera, Karen Martins, Melissa Gregory, Ivan Zuluaga, Alfonso Sucerquia Hernández, José Forero Mejía, David Boyle, Gary S. Firestein, Gary L. Simon.

**Validation:** Aileen Y. Chang, Sarah R. Tritsch, Samuel Simmens, Richard L. Amdur, Alfonso Sucerquia Hernández, José Forero Mejía.

**Visualization:** Aileen Y. Chang, Samuel Simmens, Richard L. Amdur, José Forero Mejía, Gary S. Firestein.

**Writing – original draft:** Aileen Y. Chang, Samuel Simmens, Gary S. Firestein, Gary L. Simon.

**Writing – review & editing:** Aileen Y. Chang, Sarah R. Tritsch, Carlos Andres Herrera Gomez, Liliana Encinales, Andres Cadena Bonfanti, Wendy Rosales, Evelyn Mendoza-Torres, Samuel Simmens, Richard L. Amdur, Christopher N. Mores, Paige Fierbaugh, Carlos Alberto Perez Hernandez, Geraldine Avendaño, Paula Bruges Silvera, Yerlenis Galvis Crespo, Alberto David Cabana Jimenez, Jennifer Carolina Martinez Zapata, Dennys Jimenez, Estefanie Osorio-Llanes, Jairo Castellar-Lopez, Karol Suchowiecki, Karen Martins, Melissa Gregory, Ivan Zuluaga, Abigale Proctor, Alfonso Sucerquia Hernández, Leandro Sierra, Maria Villanueva Colpas, Juan Carlos Perez Hernandez, Andres Alberto Figueroa Quast, Joaquin Andres Calderon De Barros, José Forero Mejía, Johan Penagos Ruiz, David Boyle, Gary S. Firestein, Gary L. Simon.

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
