## [Decision Letter · Decision Letter 0]

18 Dec 2023

PONE-D-23-30658Cytokine and T cell responses in post-chikungunya viral arthritis: A cross-sectional studyPLOS ONE

Dear Dr. Chang,

Thank you for submitting your manuscript to PLOS ONE. After careful consideration, we feel that it has merit but does not fully meet PLOS ONE’s publication criteria as it currently stands. Therefore, we invite you to submit a revised version of the manuscript that addresses the points raised during the review process. Please simplify some part of the presentation and carrefully defined the arthritis flare as an example. Please submit your revised manuscript by Feb 01 2024 11:59PM. If you will need more time than this to complete your revisions, please reply to this message or contact the journal office at plosone@plos.org . Please include the following items when submitting your revised manuscript:A rebuttal letter that responds to each point raised by the academic editor and reviewer(s). You should upload this letter as a separate file labeled 'Response to Reviewers'.A marked-up copy of your manuscript that highlights changes made to the original version. You should upload this as a separate file labeled 'Revised Manuscript with Track Changes'.An unmarked version of your revised paper without tracked changes. You should upload this as a separate file labeled 'Manuscript'.If applicable, we recommend that you deposit your laboratory protocols in protocols.io to enhance the reproducibility of your results. Protocols.io assigns your protocol its own identifier (DOI) so that it can be cited independently in the future. For instructions see: https://journals.plos.org/plosone/s/submission-guidelines#loc-laboratory-protocols . Additionally, PLOS ONE offers an option for publishing peer-reviewed Lab Protocol articles, which describe protocols hosted on protocols.io. Read more information on sharing protocols at https://plos.org/protocols?utm_medium=editorial-email&utm_source=authorletters&utm_campaign=protocols .

We look forward to receiving your revised manuscript.

Kind regards,

Pierre Roques, Ph.D.

Academic Editor

PLOS ONE

“I have read the journal's policy and the authors of this manuscript have the following competing interests: AYC reports consulting for Valneva. RLA reports stock ownership of Abbvie, Pfizer, Johnson & Johnson, & Briston Myers Squibb. GSF reports consulting for Eli Lilly. Other authors report no commercial interest or conflicts of interest in relation to this work.”

Please confirm that this does not alter your adherence to all PLOS ONE policies on sharing data and materials, by including the following statement: "This does not alter our adherence to PLOS ONE policies on sharing data and materials.” (as detailed online in our guide for authors http://journals.plos.org/plosone/s/competing-interests ). If there are restrictions on sharing of data and/or materials, please state these. Please note that we cannot proceed with consideration of your article until this information has been declared.

4. Please include captions for your Supporting Information files at the end of your manuscript, and update any in-text citations to match accordingly. Please see our Supporting Information guidelines for more information: http://journals.plos.org/plosone/s/supporting-information .

Reviewers' comments:

Reviewer's Responses to Questions

**Comments to the Author**

1. Is the manuscript technically sound, and do the data support the conclusions?

Reviewer #1: Yes

2. Has the statistical analysis been performed appropriately and rigorously? 

Reviewer #1: Yes

3. Have the authors made all data underlying the findings in their manuscript fully available?

Reviewer #1: Yes

4. Is the manuscript presented in an intelligible fashion and written in standard English?

Reviewer #1: Yes

5. Review Comments to the Author

Reviewer #1: The manuscript: “Cytokine and T cell responses in post-chikungunya viral arthritis: A cross-sectional study”. The main objective of this study was “to define the relationship between chronic chikungunya post-viral arthritis disease severity, cytokine response and T cell subsets in order to identify potential targets for therapy”, for which were included participants with chikungunya arthritis.

The methods used such as type of study, study population and the questionaries used for the evaluation of the severity were adequate. The data, the analysis and interpreted of the these were accurately and adequate for addressing the research question, also were enough to draw conclusions. The results are presented in adequate, and the discussion was concrete and well supported.

The authors provided enough information to validate the study.

The study has a limitation with relation to that is not included healthy control, however, the results obtained comparing different groups according to the severity of the disease drive valid results.

I think that it is well.

Other observations

In the “Participants”, “Study Design" and “Statistical analyses” sections, the authors described the study population ("participants"). On the other hand, in “Statistical analyses” section, was describe the way in which they were recruited.

I suggest that the different descriptions of the participants and the way of recruitment be included in a single section. This adjustment would allow unify this concept in one single section.

In the section: “Statistical analyses”, the authors included the study design.

I suggest, if the authors are agreed, that the study design be included in the section “Study Design".

I suggest, if it is possible, describe the sensibility and specificity of IgG antibody immunofluorescence (EUROIMMUN) test.

In row 333 the word "also" is not appropriate with the finding

In row 343 the following phrase: “the deficiency of which promotes Treg accumulation in inflamed issue(17).”, could be more appropriate for the discussion..

In the text and in the tables is used the concepts, "Arthritis-Flare" and "arthritis relapse". I suggest that be used only single of these, for example "Arthritis-Flare". It

6. PLOS authors have the option to publish the peer review history of their article (what does this mean? ). If published, this will include your full peer review and any attached files.

**Do you want your identity to be public for this peer review?** For information about this choice, including consent withdrawal, please see our Privacy Policy .

Reviewer #1: No

---

## [Author Response · Author response to Decision Letter 0]

10 Jan 2024

PONE-D-23-30658

Cytokine and T cell responses in post-chikungunya viral arthritis: A cross-sectional study

PLOS ONE 

Response to Reviewers:

Academic Review:

Please simplify some part of the presentation and carefully defined the arthritis flare as an example.

Thank you for your review. We have unified the participant recruitment description into the study design section as suggested. We have clearly defined arthritis flare and removed references to arthritis relapses as suggested.

Thank you, we confirm adherence to the PLOS ONE style requirement.

Our de-identified trial dataset is limited to a single excel that we will upload as a supplementary file for open access.

“I have read the journal's policy and the authors of this manuscript have the following competing interests: AYC reports consulting for Valneva. RLA reports stock ownership of Abbvie, Pfizer, Johnson & Johnson, & Briston Myers Squibb. GSF reports consulting for Eli Lilly. Other authors report no commercial interest or conflicts of interest in relation to this work.”

Please confirm that this does not alter your adherence to all PLOS ONE policies on sharing data and materials, by including the following statement: "This does not alter our adherence to PLOS ONE policies on sharing data and materials.” (as detailed online in our guide for authors http://journals.plos.org/plosone/s/competing-interests ). If there are restrictions on sharing of data and/or materials, please state these. Please note that we cannot proceed with consideration of your article until this information has been declared.

We confirm this statement and have added this to our cover letter, “This does not alter our adherence to PLOS ONE policies on sharing data and materials”. 

4. Please include captions for your Supporting Information files at the end of your manuscript, and update any in-text citations to match accordingly. Please see our Supporting Information guidelines for more information: http://journals.plos.org/plosone/s/supporting-information .

We have added a caption while adding the dataset as a supplementary file as per the naming guidelines provided.

The references have been reviewed.

Reviewer #1: The manuscript: “Cytokine and T cell responses in post-chikungunya viral arthritis: A cross-sectional study”. The main objective of this study was “to define the relationship between chronic chikungunya post-viral arthritis disease severity, cytokine response and T cell subsets in order to identify potential targets for therapy”, for which were included participants with chikungunya arthritis.

The methods used such as type of study, study population and the questionaries used for the evaluation of the severity were adequate. The data, the analysis and interpreted of the these were accurately and adequate for addressing the research question, also were enough to draw conclusions. The results are presented in adequate, and the discussion was concrete and well supported.

The authors provided enough information to validate the study.

The study has a limitation with relation to that is not included healthy control, however, the results obtained comparing different groups according to the severity of the disease drive valid results.

I think that it is well.

Other observations

In the “Participants”, “Study Design" and “Statistical analyses” sections, the authors described the study population ("participants"). On the other hand, in “Statistical analyses” section, was describe the way in which they were recruited.

I suggest that the different descriptions of the participants and the way of recruitment be included in a single section. This adjustment would allow unify this concept in one single section. In the section: “Statistical analyses”, the authors included the study design. I suggest, if the authors are agreed, that the study design be included in the section “Study Design".

Thank you for this suggestion, the all the participant recruitment information was moved into the ‘Study Design’ section.

I suggest, if it is possible, describe the sensibility and specificity of IgG antibody immunofluorescence (EUROIMMUN) test.

This information has been added to the ‘Chikungunya IgG Immunofluorescence’ section, “This assay has a sensitivity of 97% and specificity of 96%.”

In row 333 the word "also" is not appropriate with the finding

Thank you, ‘also’ has been removed.

In row 343 the following phrase: “the deficiency of which promotes Treg accumulation in inflamed issue(17).”, could be more appropriate for the discussion..

This comment has been moved to the discussion.

In the text and in the tables is used the concepts, "Arthritis-Flare" and "arthritis relapse". I suggest that be used only single of these, for example "Arthritis-Flare". 

In the ‘Disease Activity’ section, the definition of an arthritis flare is further elaborated and all references to ‘relapses’ has been changed to ‘flares’ for improved clarity.

---

## [Decision Letter · Decision Letter 1]

23 Jan 2024

PONE-D-23-30658R1Cytokine and T cell responses in post-chikungunya viral arthritis: A cross-sectional studyPLOS ONE

Dear Dr. Chang,

Thank you for submitting your manuscript to PLOS ONE. After careful consideration, we feel that it has merit but does not fully meet PLOS ONE’s publication criteria as it currently stands. Therefore, we invite you to submit a revised version of the manuscript that addresses the points raised during the review process.

We look forward to receiving your revised manuscript.

Kind regards,

Pierre Roques, Ph.D.

Academic Editor

PLOS ONE

**Additional Editor Comments:**

As noted there is some mistake in the cohort description as a patient below or from 18 years of age were seems to have been included in the study. Please check if discarding the data from these patients (specificaly the youngest one) impact or not the statistical analysis. Please provide a new computation at least for the reviewer. If there are not modification discard these patients that are not in the described cohort and refresh all the results. If there is a modification please provide some explanation and indicate this as a major limitation of your study.

Reviewers' comments:

Reviewer's Responses to Questions

**Comments to the Author**

1. If the authors have adequately addressed your comments raised in a previous round of review and you feel that this manuscript is now acceptable for publication, you may indicate that here to bypass the “Comments to the Author” section, enter your conflict of interest statement in the “Confidential to Editor” section, and submit your "Accept" recommendation.

Reviewer #1: All comments have been addressed

2. Is the manuscript technically sound, and do the data support the conclusions?

Reviewer #1: Yes

3. Has the statistical analysis been performed appropriately and rigorously? 

Reviewer #1: Yes

4. Have the authors made all data underlying the findings in their manuscript fully available?

Reviewer #1: Yes

5. Is the manuscript presented in an intelligible fashion and written in standard English?

Reviewer #1: Yes

6. Review Comments to the Author

Reviewer #1: 1. The authors made the changes suggested in the first evaluation. However, I have a new comment that is important to clarify before publishing the manuscript:

1. In the row 117 describes the following: “Participants over 18 years of age with a history of CHIKV infection were analyzed,” but the database includes one 6-year-old participant and three 18-year-old participants. I suggest two options: 1) that this data be clarified in the manuscript, for example "Participants 18 years of age or older and one participant 6 years of age with a history of CHIKV infection were analyzed" or 2) that the age data be removed in this row, for example: "Participants with a history of CHIKV infection were analyzed", because the mean and standard deviation of the age of the participants are described in the results.

2. Is the manuscript technically sound, and do the data support the conclusions?

The methods used such as type of study, study population and the questionaries used for the evaluation of the severity were adequate. The data, the analysis and interpreted of the these were accurately and adequate for addressing the research question, also were enough to draw conclusions. The results are presented in adequate, and the discussion was concrete and well supported.

On the other hand, the authors provided enough information to validate the study The study has a limitation with relation to that is not included healthy control, however the results obtained comparing different groups according to the severity of the disease drive valid results.

3. Has the statistical analysis been performed appropriately and rigorously?

The statistical programs used in this study were adequate and the statistical analysis was developed in an appropriate and rigorous manner for the types of variables included in the study, which allowed us to respond to the objective of the study.

4. Have the authors made all data underlying the findings in their manuscript fully available?

The authors made all data underlying the findings in their manuscript fully available

5. Is the manuscript presented in an intelligible fashion and written in standard English?

The manuscript was presented in an intelligible fashion and written in standard English.

7. PLOS authors have the option to publish the peer review history of their article (what does this mean? ). If published, this will include your full peer review and any attached files.

**Do you want your identity to be public for this peer review?** For information about this choice, including consent withdrawal, please see our Privacy Policy .

Reviewer #1: No

---

## [Author Response · Author response to Decision Letter 1]

26 Jan 2024

Academic Review:

As noted there is some mistake in the cohort description as a patient below or from 18 years of age were seems to have been included in the study. Please check if discarding the data from these patients (specifically the youngest one) impact or not the statistical analysis. Please provide a new computation at least for the reviewer. If there are not modification discard these patients that are not in the described cohort and refresh all the results. If there is a modification please provide some explanation and indicate this as a major limitation of your study.

Thank you very much for catching this error. I have checked our source documents and participant 75 is not 6 years of age but rather 59 years of age. There was a typo when transcribing from the source documents and the age was confirmed with their date of birth from their identification card. This change has been made in the dataset and the mean ages edited accordingly in Table 1. Furthermore, the text at row 117 has been edited to reflect that participants were age 18 and older in this study. 

Reviewer #1: 

1. The authors made the changes suggested in the first evaluation. However, I have a new comment that is important to clarify before publishing the manuscript:

1. In the row 117 describes the following: “Participants over 18 years of age with a history of CHIKV infection were analyzed,” but the database includes one 6-year-old participant and three 18-year-old participants. I suggest two options: 1) that this data be clarified in the manuscript, for example "Participants 18 years of age or older and one participant 6 years of age with a history of CHIKV infection were analyzed" or 2) that the age data be removed in this row, for example: "Participants with a history of CHIKV infection were analyzed", because the mean and standard deviation of the age of the participants are described in the results.

Thank you very much. Please see the response from above.

---

## [Decision Letter · Decision Letter 2]

13 Feb 2024

Cytokine and T cell responses in post-chikungunya viral arthritis: A cross-sectional study

PONE-D-23-30658R2

Dear Dr. Chang,

We’re pleased to inform you that your manuscript has been judged scientifically suitable for publication and will be formally accepted for publication once it meets all outstanding technical requirements.

An invoice for payment will follow shortly after the formal acceptance. To ensure an efficient process, please log into Editorial Manager at http://www.editorialmanager.com/pone/ , click the 'Update My Information' link at the top of the page, and double check that your user information is up-to-date. If you have any billing related questions, please contact our Author Billing department directly at authorbilling@plos.org .

If your institution or institutions have a press office, please notify them about your upcoming paper to help maximize its impact. If they’ll be preparing press materials, please inform our press team as soon as possible -- no later than 48 hours after receiving the formal acceptance. Your manuscript will remain under strict press embargo until 2 pm Eastern Time on the date of publication. For more information, please contact onepress@plos.org .

Kind regards,

Pierre Roques, Ph.D.

Academic Editor

PLOS ONE

Additional Editor Comments (optional):

Reviewers' comments:

Reviewer's Responses to Questions

**Comments to the Author**

1. If the authors have adequately addressed your comments raised in a previous round of review and you feel that this manuscript is now acceptable for publication, you may indicate that here to bypass the “Comments to the Author” section, enter your conflict of interest statement in the “Confidential to Editor” section, and submit your "Accept" recommendation.

Reviewer #1: All comments have been addressed

2. Is the manuscript technically sound, and do the data support the conclusions?

Reviewer #1: Yes

3. Has the statistical analysis been performed appropriately and rigorously? 

Reviewer #1: Yes

4. Have the authors made all data underlying the findings in their manuscript fully available?

Reviewer #1: Yes

5. Is the manuscript presented in an intelligible fashion and written in standard English?

Reviewer #1: Yes

6. Review Comments to the Author

Reviewer #1: The methods used such as type of study, study population and the questionaries used for the evaluation of the severity were adequate. The data, the analysis and interpreted of the these were accurately and adequate for addressing the research question, also were enough to draw conclusions.

The results are presented in adequate, and the discussion was concrete and well supported. On the other hand, the authors provided enough information to validate the study The study has a limitation with relation to that is not included healthy control, however the results obtained comparing different groups according to the severity of the disease drive valid results.

The statistical programs used in this study were adequate and the statistical analysis was developed in an appropriate and rigorous manner for the types of variables included in the study, which allowed us to respond to the objective of the study.

The authors made all data underlying the findings in their manuscript fully available.

The manuscript is presented in an intelligible fashion and written in standard English.

7. PLOS authors have the option to publish the peer review history of their article (what does this mean? ). If published, this will include your full peer review and any attached files.

**Do you want your identity to be public for this peer review?** For information about this choice, including consent withdrawal, please see our Privacy Policy .

Reviewer #1: No

---

## [Editor Report · Acceptance letter]

11 Mar 2024

PONE-D-23-30658R2 

PLOS ONE

Dear Dr. Chang, 

I'm pleased to inform you that your manuscript has been deemed suitable for publication in PLOS ONE. Congratulations! Your manuscript is now being handed over to our production team.

Lastly, if your institution or institutions have a press office, please let them know about your upcoming paper now to help maximize its impact. If they'll be preparing press materials, please inform our press team within the next 48 hours. Your manuscript will remain under strict press embargo until 2 pm Eastern Time on the date of publication. For more information, please contact onepress@plos.org .

If we can help with anything else, please email us at customercare@plos.org .

Kind regards, 

on behalf of

Dr. Pierre Roques 

Academic Editor

PLOS ONE